

# Understanding the combined mental health impacts
# of flooding and COVID-19 in Hue City, Central Vietnam

Thi Dieu My Pham[1], Paul Hudson[2], Annegret H Thieken[1], Philip Bubeck[1]

[1]Institute of Environmental Science and Geography, University of Potsdam
     [2]Department of Environment and Geography, University of York, York, United Kingdom

*Correspondence to*: Thi Dieu My Pham (thi.dieu.my.pham@uni-potsdam.de)

**Abstract**

Experiencing severe flooding tends to negatively impact mental health, creating a significant public health issue.

Moreover, extreme events can co-occur, magnifying potential impacts. Understanding the combined mental health impacts of floods and COVID-19 is a research gap we addressed by conducting 400 face-to-face surveys in October 2023 in Hue City, Vietnam, where residents faced simultaneous flooding and COVID-19 in 2020.

The respondents' mental health was assessed using the Kessler psychological distress scale (K6), revealing that 20% of the respondents experienced mental health distress, while 80% did not report such distress. Binary logistic

regression models demonstrated that among twelve flood stressors, facing 'livelihood difficulties', 'seeing dead human bodies', and 'being rescued' relate significantly to mental distress. Meanwhile, 'impacts on individual health' and 'interrupted education' are the two significant COVID-19 stressors. These five factors stay significant when combined. Additionally, a multivariable regression model revealed the combined effects of flood and COVID-19 when comparing the ORs of four groups ranging from 'No flood stress & No Covid stress' to 'Flood stress & Covid

stress'. The effect size is largest for those who experienced flood and COVID-19 impacts, followed by those who suffered only floods and those who faced only COVID-19, with the smallest effect size.

These findings underline the need to address public health problems caused by multiple risks, which is still a significant gap in developing countries. Furthermore, psychological impacts could be reduced by providing additional support to at-risk communities, like managing human remains, rehearsing evacuation plans, preventing school

closures, and setting up public health infrastructure for psychological assistance.

Keywords: Mental Health, Flood, COVID-19, Combined Impacts, Multi-risk Management



## 1 Introduction

Among climate-related disasters, flooding is the most disruptive and costly disaster experienced worldwide (IPCC, 2021; Liu et al., 2024). From 1990 to 2022, 4713 flood events were recorded in 168 countries, killing 218,353 people, of which the region of Southeast Asia has a high proportion of deaths with 33%, and causing an economic loss of 1.3 trillion USD (Liu et al., 2024). Besides causing significant numbers of human casualties, floods are causing damage to infrastructure, property, and agricultural livelihoods (UNDRR, 2020). There has been a global increase in the

number of affected people and assets in flood-prone areas (Moel et al., 2015; UNISDR, 2011), which, when combined with a projected increase in the magnitude and frequency of hazards from climate change (Seneviratne et al., 2021), indicates growing threats (IPCC, 2023).

In addition to the monetary and physical flood damage, there is significant evidence of both short-term and long-term flood impacts on psychological well-being (Arshad et al., 2020; Paranjothy et al., 2011; Abass et al., 2022; Arshad et

al., 2020; Butler et al., Adger, 2018; Tunstall et al., 2006; Hudson et al., 2019). Long-lasting mental health disorders such as post-traumatic stress disorder (PTSD), anxiety and depression have higher incident rates post-flood (Stanke et al.,2012; Zenker et al., 2024). IPCC (2023) reports also project that flooding in Asia will significantly impact human well-being. Therefore, the effects of flooding on mental health are being increasingly recognized and must be proactively integrated into risk management, as it has been largely neglected (Berry et al., 2018; Gifford & Gifford,

2016). Furthermore, the mental health impacts of flooding have been studied in high-income countries; it is crucial to recognize that flooding also has profound mental health effects on poor people in low-income countries as well (Asim et al., 2022).

The need to understand the public health consequences of flooding is complicated by the observation that people can face or experience multiple hazards that may cause cumulative impacts rather than only floods. A prominent situation

was the COVID-19 pandemic, which occurred simultaneously with other disasters in many places in 2020. It was a global crisis in health systems and economies worldwide, occurring with socio-environmental changes, causing compound effects, emphasizing the urgent need for an integrated and intersectoral approach to understanding and addressing risks and impacts of such crises on the most vulnerable populations (UNDRR, 2019).

The co-occurrence of floods and COVID-19 is a typical multi-hazard event (Gill et al., 2022; Simonovic et al., 2021).

While the flood impacts on mental health are considerable and long-lasting, a systematic review of health sector responses to the coincidence of disasters and COVID-19 are underreported and under-evaluated (Ogunbode et al., 2019; Sedighi et al., 2021). Compared to 478 studies on the prevalence of PTSD among flood survivors reviewed by Golitaleb et al. (2022) from 2015 to 2021, a few examples highlight cumulative impact assessments for multiple risks in combination with floods. Sohrabizadeh et al. (2021) reviewed and found thirteen studies addressing the co-

occurrence of COVID-19 and natural hazards in general. Seven reported the simultaneous occurrence of COVID-19 and climatic events like floods, hurricanes, tornadoes, and southwest monsoons in South Asian countries. Only two of these seven articles worked on the health consequences of co-occurring floods and the pandemic, including Vikas (2020) and Guo et al. (2020). Vikas (2020) studied the potential risks when monsoons in Southeast Asia trigger floods during COVID-19, which may prevent the healthcare system from being prepared for the disease. The author

emphasized the necessity of the mitigation action plan against COVID-19 during monsoons. Meanwhile, Guo et al. (2020) mentioned possible flood impacts during the pandemic in China. They drew attention to the necessity of urgent



actions to prevent the health and livelihood consequences of flooding during the COVID-19 pandemic. However, none of these articles examines the combined impacts of floods and pandemics on mental health.

We conducted a further search on the Web of Science and PubMed databases using the strings: 'flood', 'COVID-19', AND 'mental health' in December 2023 (see Appendix 1); we found 52 articles in the Web of Science and 55 articles in PubMed. By scanning the titles and abstracts, the duplicated results were removed. There were 18 relevant articles, but only seven recently explored the compounding impacts of flood and COVID-19 on affected people's mental health (Agyapong et al., 2021; Podubinski & Glenister, 2021; Rocha et al., 2021; Agyapong et al., 2022; Callender et al., 2022; Izumi & Shaw, 2022; Liang et al., 2023), meaning that our research will contribute significantly to this field. Agyapong et al. (2021) focused on the cumulative impacts on mental health after wildfire, flooding and COVID-19 on Fort McMurray school board staff and other employees in Canada, and they concluded that affected groups suffered psychological morbidity differently. Podubinski and Glenister (2021) provided insights into the mental health of Australian workers during the initial COVID-19 outbreak, with an additional focus on whether previous disaster exposure and effects from that disaster are risk factors for increased psychological distress. They found that higher stress symptoms were associated with having disaster impacts, added to COVID-19. In the meantime, Rocha et al. (2021) aimed to address the effects of natural disasters on the mental health of Filipinos during the COVID-19 pandemic, with the conclusion that the simultaneous existence of natural disasters and the pandemic has caused devastating and detrimental effects on the mental health of Filipinos. Also, Agyapong et al. (2022) found that cumulative trauma from multiple natural disasters, including COVID-19, has increased the mental health burden on residents of Fort McMurray in Canada. Callender et al. (2022) assessed the economic and mental health impacts of COVID-19 in the presence of previous exposure to flood events caused by Hurricane Harvey in the US. They found that multiple crises can jointly and cumulatively shape health and well-being. Izumi and Shaw (2022) examined the effects of COVID-19 and natural hazards, including floods, in countries like India, Japan, the Philippines, and the USA, concluding that the confluence of COVID-19 and natural hazards caused compounded impacts and challenges to mental health. Lastly, Liang et al. (2023) identified the latent profiles of psychological status and acceptance of change among Henan residents in China who have been cumulatively exposed to floods and the COVID-19 pandemic. They found that the additive effects of the floods with COVID-19 have a predictive effect on psychological state. From this literature review, we found that the research on the cumulative impacts of co-occurring disasters is still limited, and the seven mentioned works used different methods and scales for assessment; none used the K6 scale as a screening tool. Therefore, we want to contribute novelty to these research gaps by surveying the cumulative impacts of flooding and COVID-19 on the mental health of at-risk communities in Hue City. The K6 scale appears to be a proper choice for comparisons across countries and regions, as it is a widely used screening scale for mental health issues in the general adult population (Kessler et al., 2002).

Our research question is: Does the co-occurrence of floods and COVID-19 have combined impacts on mental health? We used the K6 scale distress to examine and answer the research question in the case study conducted in Hue City (which used to be Thua Thien Hue province before January 13th, 2025), Vietnam. This city faced COVID-19's first wave in January 2020 and the second wave in July 2020, leading to social isolation until December 2020, with hard lockdowns and restrictions in many communities (Nguyen et al., 2021). In October 2020, Hue City faced typhoons, floods and landslides. Therefore, local people suffered uncertainty, fear of infection, distress, grief, and loneliness (Nguyen et al., 2021).



The research results draw attention to the combined mental health impacts of multiple disasters and provide a better understanding of how these have become increasingly common in recent years. Our findings give important insights to direct relevant stakeholders to preparedness, address public health issues in this context, and identify risk factors that exacerbate psychological distress in order to offer thoughtful solutions to mitigate impacts.

## 110     2 Case study area

Vietnam is a developing country in Asia (UN, 2024). It is located in the tropical region and is one of the most vulnerable countries to climate change impacts worldwide, including hydro-meteorological hazards such as severe storms, cyclones, typhoons, floods, and landslides (CFE-DM, 2021). About 70% of Vietnam residents live in coastal communities and, as such, are highly exposed to intensifying storms and floods (CFE-DM, 2021). Floods cause this

country's most significant economic loss, accounting for 97% of average annual losses from hazards (WB & ADB, 2021). Especially in 2020, when a storm triggered massive floods in Central Vietnam, people suffered heavy losses, including damage to people, shelters and property (IFRC, 2022). 357 people were killed or missing, and 876 were injured. Storms and floods that occur in a row are considered the most terrible disasters affecting Central Vietnam in the past 100 years (IFRC, 2022).

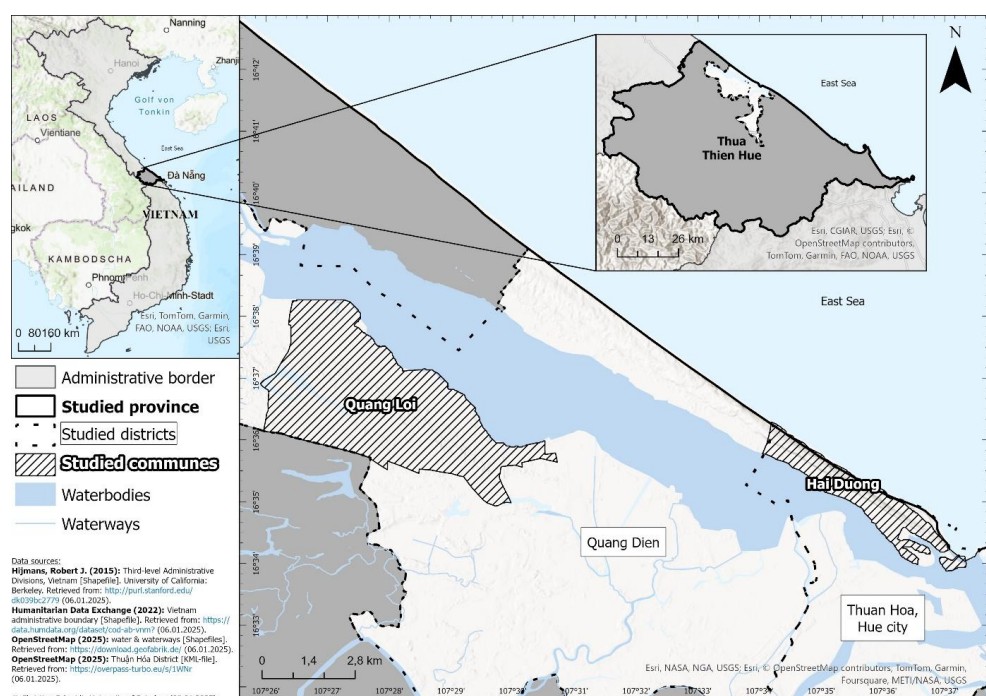

*Figure 1. Study sites*

Located in coastal Central Vietnam (Fig.1), Hue City is especially prone to frequent and severe coastal, pluvial, and fluvial flooding due to its location, the river system and tropical monsoon climate, with a rainy season occurrence from August to December (Tran et al., 2008). Additionally, the city suffers from typhoons that frequently make

landfall in this area with massive rain during the rainy season (Tran et al., 2008). This means Hue people have been



subjected to flood risk physically or mentally, as analyzed by Sett et al. (2024). As far as flood risk is concerned, 2020 is an example of this vulnerable situation. The city suffered tropical cyclones, bringing massive and unusual floods in October that lasted about one month and triggering landslides in many places. Widespread flooding and landslides caused severe damage and loss of life in Hue City in 2020, with 31 people dead and 11 people missing and a total

economic damage of 45 million USD (Van Dinh, 2020). Tran and Vilas (2011), Hudson et al. (2021) and French et al. (2019) stated that flood impacts on mental and physical health may be long-lasting and affect households for years after the flood events, especially in areas with frequent flooding like Hue City.

In addition to severe floods in 2020, the COVID-19 pandemic brought many difficulties to the public health, research, and medical communities worldwide, including Vietnam (WHO, 2020a). According to WHO (2022b),  455 million

cases and over 6 million deaths were recorded by mid-March 2022. In Vietnam in general and Hue City in particular, COVID-19 cases increased in January 2020, and people faced social isolation from February to December 2020 (Nguyen et al., 2021) caused by hard lockdowns and restrictions in many communities. This situation resulted in the cessation of livelihood activities, the closure of schools and offices, family separations, and travel restrictions. Vietnam controlled the first wave in April 2020, but a second wave of infections occurred in July 2020. Immediately

after managing the second wave of COVID-19, Hue City encountered typhoons, floods, and landslides from October 10th until the end of the month. During and after these disaster events, social distancing measures were enforced until December 2020 (Nguyen et al., 2021), causing negative impacts on the psychological state of those affected (Peteet, 2020).

Our surveys on the combined effects of floods and COVID-19 were conducted in two coastal communes in Hue City,

namely Hai Duong and Quang Loi (see Fig.1).  Due to their location, these communes face flood and storm hazards yearly. In 2020, both municipalities were also affected by COVID-19 and the nationwide measures to contain the pandemic. Hai Duong commune has belonged administratively to Hue City since 2021. It is located along the coastline between the lagoon and the East Sea, with a total area of 9,693 km² and a population of 8,190 (as of 2021). The residents in this commune face coastal erosion, storms, floods, and drought annually and are also severely affected by

the extreme floods of 2020. Quang Loi commune is located in the northeast of Quang Dien district, along the lagoon, and occupies 32.32 km² with a population of 6,247 people (Tran et al., 2021).

### 3 Methods

#### 3.1 Data collection

KOBO Toolbox was used to conduct 400 face-to-face surveys in the two coastal communes from October 6 to 30th,

2023, including Hai Duong and Quang Loi. The questionnaires focused on psychological distress, personality traits, flood impacts, and COVID-19 impacts on affected people. Regarding the sampling method, the survey team went through the commune and randomly selected every third household along the village lanes for a face-to-face interview. If a household denied the interview, the respective household was skipped, and the household next to it was chosen. The trained and experienced local enumerators continued until 200 interviews per commune were completed. All of

the 400 questionnaires were collected for analysis.



### 3.2 Measuring psychological distress (K6)

The respondents' mental health was assessed using the Kessler Psychological Distress Scale (K6), a widely used brief screening scale to assess non-specific psychological distress (Kessler et al., 2002). The K6 asks respondents how often six symptoms occurred in the 30 days before the interview: 1) feeling nervous, 2) hopeless, 3) restlessness or fidgety, 4) feeling worthless, 5) feeling depressed, and 6) that everything was an effort (Kessler et al., 2002). The answers for each question range from 0 to 4, assigning 'none of the time', 'a little of the time', "some of the time", 'most of the time', or 'all of the time', respectively. From that, the total score of the six symptoms ranges from zero to 24, with a higher score indicating a higher mental health distress. The K6 is widely used and has demonstrated promising results in various contexts of population-based health surveys (Prochaska, 2012) and assessed by the WHO (Kessler et al., 2010). For example, it was employed to screen for psychological distress in the Australian National Survey of Mental Health and Well-Being (Furukawa et al., 2003), to examine serious mental illness in the general population (Kessler et al., (2003), to assess any disorder in the community mental health surveys as a cautionary note (Veldhuizen et al.,(2007), to serve as a screening instrument for Iranian University Students (Dadfar et al., 2018), and to study Indian Americans regarding the severity of mood disorders (Mitchell & Beals, 2011). The K6 is confirmed to be a reliable tool for screening adult psychological distress with panic disorder, generalized anxiety disorder, bipolar disorder, and schizophrenia (Umucu et al., 2022; Wijeratne et al., 2011). In Vietnam, Kawakami et al. (2020) researched internal consistency reliability, construct validity, and item response characteristics of the Kessler 6 scale among hospital nurses. They concluded that the K6 Vietnamese version is a reliable and valid instrument to measure psychological distress for their targeted group.

To classify the psychological distress level of respondents, we use cut-off points derived from the literature (Prochaska et al., 2012). The standard cut-off points are <5 for 'no mental health distress', $\geq 5$ and $< 13$ for 'moderate mental distress', and $\geq 13$ for 'severe mental health distress' (Prochaska et al., 2012). Prochaska et al. (2012) determined the validity of the standard cut-off points for a sample in the US, which analyzed 50,880 adult participants in a 2007 California Health Interview Survey. Their findings indicate that the optimal cut-off points identified through the receiver operating characteristic curve analysis are consistent across various ethnic and racial groups. They advocate for the expansion and analysis of the K6 scale to measure and examine the associated factors with moderate mental distress. Min and Lee (2015) also justified that these cut-off points are proposed for Korean seniors and that the K6 is a valid and reliable screening tool.

### 3.3 Variable selection

Our study has four variable groups: 1) dependent variables, 2) flood stressors, 3) COVID-19 stressors and 4) contextual variables. The variable domains were selected based on the literature review and the local context, reflecting the specific disaster situation in 2020 in Hue City. Regarding flood stressors, we considered what affected people suffered and their safety. As stated by Lee et al. (2020) and Du et al. (2010), adverse mental health status caused by disasters is generally due to a combination of physical health problems, financial losses, and community or social disruption; hence, we include the main factors comprising home damage, livelihood difficulties, food and water shortage, unsanitary conditions and disrupted medicine/medical care. These factors were asked in separate questions in the flood section. For people's safety during the flood, as mentioned by McKenzie et al. (2022), being flooded may cause physical injury, property damage, evacuation or resettlement, and severe disruption, all of which are expected



consequences referred to as primary stressors. In the specific case of Hue City, the local context that affects mental health was mentioned in the research by Sett et al. (2024), that mental health impacts could directly stem from seeing dead bodies, being injured and being infected by diseases due to poor sanitation conditions. For a detailed description of all variables included in this study and statistical models, please see Appendix 2.

Regarding COVID-19 impacts, questions were adopted from the survey of CSO (2020) on the social implications of
the pandemic. Our questions focused on COVID-19 information access, individual diagnosis of COVID-19, and the effects of COVID-19 on jobs/livelihoods and financial obligations, health, social ties, interrupted education, household stress, and home violence. We also asked about the impacts of restrictions during the pandemic. Some other researchers included these factors in their studies as well; for example, Xiong et al. (2020), Mathew (2021), and Wiedemann et al. (2022) considered health status, married status, age, household income, and quarantine status.
Mathew (2021) and Chen et al. (2020) also added fear of infection as a risk factor. Chen et al. (2020) included 'study online' in their risk factor analysis. Béland et al. (2021) and Piquero et al. (2021) examined family stress and domestic violence during the COVID-19 outbreak, especially.

Contextual variables typically include sociodemographic factors associated with adverse mental health impacts after the flood. Depending on the research focus, these variables could be age, gender, educational status, current work
status, annual household income, and housing status (French et al., 2019; Lee et al., 2020; Fitzgerald et al., 2020; Adams & Nyantakyi-Frimpong, 2021; Asim et al., 2022; Gousse-Lessard et al., 2023; Graham, White, Cotton, & McManus, 2019). In our study, we ran single binary regression models for each variable. We found that current work status, average household income, and housing status do not affect respondents' mental health status. Also, to compare flood and COVID-19 aspects, we chose age, gender, and education as control variables.

**3.4 Statistical methods**

After selecting variable groups, descriptive statistics were used to provide an overview of the demographics and characteristics of the sample. Then, mental health prevalence was investigated to see the percentage of respondents facing mental health distress using the K6 scale score and the above cut-off points (Kessler et al., 2002). Classifying mental health status includes three levels: no mental health distress, moderate mental distress and severe mental health
distress, and presented as a category variable. A new dummy variable of psychological distress was created by combining moderate and severe mental proportions as Yes (1) and No (0) for 'No mental health' to capture the difference between groups, here, those with psychological distress and others without, and for later use in binary logistic regression models.

Later, the relation of mental health distress with the explanatory variables was tested to find out if the contextual and
risk factors influence mental health status in different ways. Three logistic regression models examined mental health risk factors (see Table 3). The model M-Flood examines the influence of twelve flood stressors on the respondents, for example, home damage, livelihood difficulties, food and water shortage, etc. Of these, five were category variables; seven others were in dummy format. To be identical, we changed the five category variables to dummy variables by creating a value of '0' (No) for the answer categories 'none', 'a little' and 'some', and a value of '1' (Yes) for 'a lot' and
'extreme'. The M-COVID-19 examined the influence of six COVID-19 stressors on the mental distress of the respondents, including the impacts on individual health, someone's health and social maintenance, etc. (Table 2). The value of '0' (No) was created from original values from 'Not at all', 'A little' and 'Somewhat', while the value of '1' (Yes) was made from 'A lot' and 'Extremely'. Building upon M-Flood and M-COVID-19, the model M-Mixed



examined the change of influence of flood and COVID-19 stressors on the psychological distress of affected people
when entered into a model simultaneously. Only significant variables in M-Flood and M-COVID-19 with p-values <
0.1 were selected for M-Mixed.

Lastly, the combined impacts of direct flood and COVID-19 stressors on mental health were investigated by grouping
respondents into different categories. Hence, the fourth model (M-Combined) was run as a multivariable logistic
regression for mental health status. To explore the combined impact of floods and COVID-19 on the mental health
distress experienced by individuals affected by both disaster and the pandemic, we adopted the grouping method by
Fitzgerald et al. (2020) who examined the cumulative flood exposure via probable depression among business owners
who were flooded and evacuated during the 2017 flood event in Australia. In our research, four groups were created:
Group (1) comprised of respondents reporting 'NO flood stress and NO Covid stress', Group (2) reporting 'Flood
stress and NO COVID stress', Group (3) reporting 'NO flood stress and COVID stress', and Group (4) reporting
'Flood stress and COVID stress'. The groups were created as shown in Fig. 2. Dummy variables were designed to
capture those respondents who faced flood or COVID stressors. Their ORs were compared to find evidence of
combined impacts.

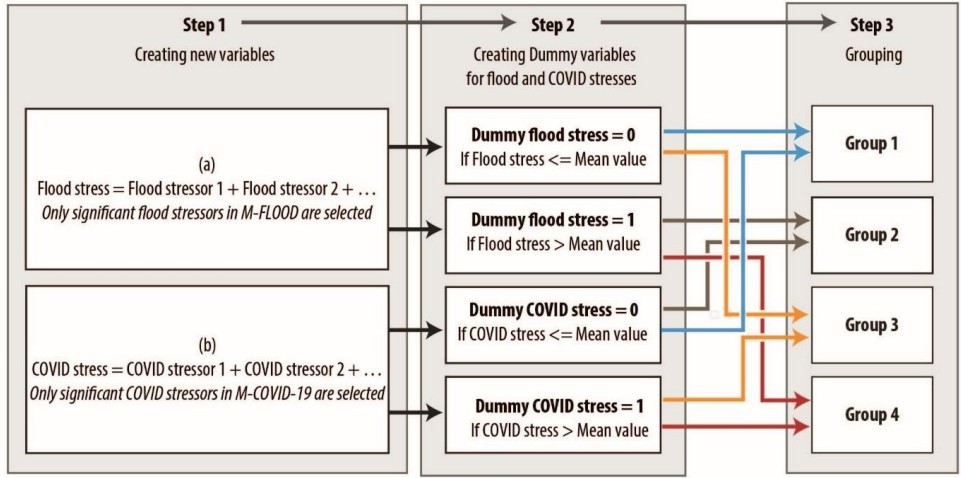

*Figure 2. Creating groups for examining the combined impacts of flood and COVID-19 on mental health distress:*
*Step 1: Summing up all significant flood stressors in M-FLOOD to create a variable (a) 'Flood stress' AND Summing up all*
*significant COVID-19 stressors in M2 to create the variable (b) 'COVID stress'*
*Step 2: Creating dummy variables for flood and COVID stressors by using the mean values of 'Flood stress' and 'COVID*
*stress'.*
*Step 3: Creating the four groups.*

Figure 2 shows that the dummy variables were created using the mean values (Cox, 2018; Cox & Schechter, 2019) to
sum up the significant stressors of both flood and COVID-19. Firstly, a general 'flood stress' was created by summing
up the three significant variables in M-Flood as (a) :

*(a) Flood stressor = Livelihood difficulties + Seeing human dead bodies + Being rescued*



From (a), we had the variable '*Flood stress'* with the values Min = 0, Max = 3 and Mean ~ 0.8. We created a dummy variable for this with '0 = No flood stress' if '*Flood stress'* ≤ 0.8 and '1= Yes, flood stress' if '*Flood stressor'* > 0.8.

Secondly, we created the general COVID-19 stressor based on the two significant variables from M-COVID-19 as (b):

*(b) COVID-19 stress = impacts on individual health + interrupted education*

From (b), we had the variable '*COVID-19 stress*' with the values Min = 0, Max = 2, and Mean ~ 0.3. Then, we created a dummy variable with '0 = No COVID stress' if '*COVID-19 stress'* ≤ 0.3 and '1= Yes, COVID stress' if '*COVID-19 stressor'* > 0.3. In the next step, we used the two new dummy variables to group them into four: *(1) NO flood stress*

*and NO COVID stress, (2) Flood stress and NO COVID stress, (3) NO flood stress and COVID stress, and (4) Flood stress and COVID stress.*

All regression models report effect sizes as ORs with a 95% confidence interval for bootstrapping. The OR represents the likelihood of an outcome occurring due to a specific exposure compared to the possibility of the outcome occurring without that exposure (Szumilas, 2010). The odds ratio can also be used to justify whether a particular risk factor takes

effect for a specific outcome and to compare the influence of several risk factors for that outcome (Szumilas, 2010). An OR of 1 means that the risk factor does not affect the odds of the outcome occurring, ceteris paribus; an OR>1 means that the risk factor is associated with higher odds of the outcome occurring; and an OR<1 implies that a risk factor is associated with lower odds of outcome occurring.

**4 Results**

**4.1 Sample Characteristics**

As demonstrated in Table 1, respondents (n = 400) comprise almost equal numbers of women (53.2%) and men (46.8%). Overall, the gender proportion matches well with the province's population (TTH Statistic Office, 2022), which shows that females make up 50.1% of the population and males 49.9%. The mean age of the sample is 49 years, with the youngest respondent being 17 years old and the oldest being 87 years old. Also, for average age, our sample

belongs to the working age group, which covers 53% of the province's population (as of 2022) (TTH Statistic Office, 2022). Regarding education, 84% of our sample are literate, meaning they have at least a primary school degree. This ratio is slightly lower than the provincial average, with 89.4% of people aged 15 and above in rural areas being literate (TTH Statistic Office, 2022). In addition, the data shows a wide range of average income levels. The respondents have a wide range of average monthly incomes. The highest proportion belongs to the group with '>3-4 million

VND/month', with 14.8%. Generally, this result does not reflect the average income reported by the Statistics Office of the TTH province, which is 4.7 million VND/month/person (as of 2023) (PPC-TTH, 2023; TTH Statistic Office, 2022). The reason could be the sensitivity of the question; people usually hesitate to talk about their income, so they choose not to answer it or report a lower income.





*Table 1. Sampling characteristics*

| | Sampling | | | Province TTH Statistic Office (2022) |
|---|---|---|---|---|
| | **Female** | **Male** | **Total** | |
| **N** | 213 (53.2%) | 187 (46.8%) | 400 (100.0%) | |
| **Location** | | | | Whole province: |
| *Hai Duong* | 108 (50.7%) | 92 (49.2%) | 200 (50.0%) | Females: 50.1% |
| *Quang Loi* | 105 (49.3%) | 95 (50.8%) | 200 (50.0%) | Males: 49.9% |
| **Age** | 49 (11.461) | 49 (12.365) | 49 (11.883) | Working age: 53% Men:15-61, Women: 15-56 |
| **Education** | | | | |
| No formal education | 26.0% | | | |
| Primary school | 32.0% | | | |
| Secondary school | 21.8% | | | 89.4% of people aged 15 and above in the rural area are literate (having primary school education level upward) |
| High school | 13.0% | | | |
| Technical Diploma | 3.5% | | | |
| University | 3.7% | | | |
| **Income** | | | | |
| <500,000 VND | 3.3% | | | |
| >500,000 – 1 million VND | 1.00% | | | |
| >1 - 2 million VND | 2.8% | | | |
| >2 - 3 million VND | 10.7% | | | |
| >3 - 4 million VND | 14.8% | | | |
| >4 - 5 million VND | 9.7% | | | |
| >5 - 6 million VND | 12.00% | | | |
| >6 – 7 million VND | 9.8% | | | |
| >7 – 8 million VND | 6.2% | | | |
| >8 – 9 million VND | 6.3% | | | |
| >9 – 10 million VND | 7.00% | | | |
| >10 – 15million VND | 7.2% | | | |
| >15 - 20 million VND | 7.7% | | | |
| >20-50 million VND | 0.7% | | | |
| Don't know/no answer | 0.8% | | | 4.7 million VND/month/person |

Regarding flood risk experience, Table 2 shows a significant proportion of respondents who faced flood and pandemic impacts. More than 97% were affected by flood events in 2020. Of these, Quang Loi had more people who experienced flood events in 2020, with 98.9% of respondents. 19.5% of respondents from both communities are still burdened by this flood, with responses from 'A lot' to 'Extremely'. Quang Loi had more people suffering 'A lot' in the flood of 2020, with 55 respondents, while Hai Duong had only two. Among the three top significant flood stressors, 'livelihood difficulties' occupy the largest proportion with 56.5% in the two communities, followed by 'being rescued' and 'seeing dead human bodies' with 24.4% and 19.2%, respectively. Again, Quang Loi has a much higher percentage of these stressors than Hai Duong.

Regarding COVID-19, 41.6% of respondents were directly affected by this pandemic, meaning they tested positive for COVID-19 or were suspected of being infected but were never tested. Of these, Quang Loi commune has more people affected, with 50% of respondents, while Hai Duong has 30%. The pandemic still burdens many people in these two communes. In particular, Quang Loi has more people answering that COVID-19 still burdens them significantly (i.e., choosing the three highest answer categories). The two significant COVID-19 risk factors, like 'impacts of COVID-19 on your health' and 'interrupted education', have considerable effects on people, with



36% and 15.8%, respectively. More people in Quang Loi faced individual health problems, while more people in Hai Duong faced interrupted education constraints.


*Table 2. Number of people affected by flood and COVID-19 in 2020*

|  | **Hai Duong** | **Quang Loi** | **Total** |
|---|---|---|---|
| **Flood exposure** |  |  |  |
| **N** | **200** | **200** | **400** |
| *Affected by flood in 2020* |  |  |  |
| Yes | 136 (97.1%) | 188 (98.9%) | 324 (98.2%) |
| No | 4 (2.9%) | 2 (1.1%) | 6 (1.8%) |
| *Burden by flood 2020* |  |  |  |
| *N* | *136* | *188* | *324* |
| Extremely | 2 (1.5%) | 4 (2.1%) | 6 (1.9%) |
| A lot | 2 (1.5%) | 55 (29.3%) | 57 (17.6%) |
| Somewhat | 24 (17.6%) | 22 (11.7%) | 46 (14.2%) |
| A little | 44 (32.4%) | 28 (14.9%) | 72 (22.2%) |
| Not at all | 64 (47.1%) | 79 (42.0%) | 143 (44.1%) |
| *Significant flood stressors* |  |  |  |
| *Livelihood difficulties* |  |  |  |
| *N* | *136* | *188* | *324* |
| Yes | 59 (43%) | 124 (66%) | 183 (56.5%) |
| No | 77 (57%) | 64 (34%) | 141 (43.5%) |
| *Being rescued* |  |  |  |
| *N* | *130* | *188* | *318* |
| Yes | 19 (14.6%) | 60 (31.9%) | 79 (24.4%) |
| No | 111 (85.4%) | 128 (68.1%) | 239 (75.6%) |
| *Seeing dead human bodies* |  |  |  |
| *N* | *135* | *188* | *323* |
| Yes | 11 (8.2%) | 51 (27.1%) | 62 (19.2%) |
| No | 124 (91.8%) | 137 (72.9%) | 261 (80.8%) |
| **COVID-19 exposure** |  |  |  |
| **N** | **200 (50.0%)** | **200 (50.0%)** | **400 (100.0%)** |
| *Individual diagnosis of Covid-19* |  |  |  |
| Yes, tested and confirmed | 59 (29.5%) | 64 (32.0%) | 123 (30.8%) |
| Suspected but not tested | 7 (3.5%) | 36 (18.0%) | 43 (10.8%) |
| No | 133 (66.5%) | 99 (49.5%) | 232 (58.0%) |
| Don't know / No answer | 1 (0.5%) | 1 (0.5%) | 2 (0.5%) |
| *Significant COVID-19 stressors* |  |  |  |
| *…on your health* |  |  |  |
| *N* | *200* | *200* | *400* |
| Yes | 39 (19.5%) | 75 (37.5%) | 114 (36%) |
| No | 161(80.5%) | 125 (62.5%) | 286 (64%) |
| *…interrupted education* |  |  |  |
| *N* | *197* | *200* | *397* |
| Yes | 37 (18.8%) | 23 (11.5%) | 60 (15.8%) |
| No | 160 (81.2%) | 177 (88.5%) | 337 (84.2%) |

**4.2. Mental health status**

Figure 3 reports the descriptive results for the six symptoms of the K6 scale. Nervousness is reported most often, with 32.3% of the respondents choosing the two highest answering categories. Restlessness follows this with 10.6%. The

remaining four symptoms have a similar proportion, around 5.1-5.6% across these categories. Nervousness and restlessness are also more frequent than other symptoms in the category 'A little of the time' with 38.5% and 32%,





respectively, and more than double the least frequent symptom, hopelessness. Hopelessness is the least frequent symptom overall, with the highest percentage for 'None of the time' and the lowest number in the remaining categories.

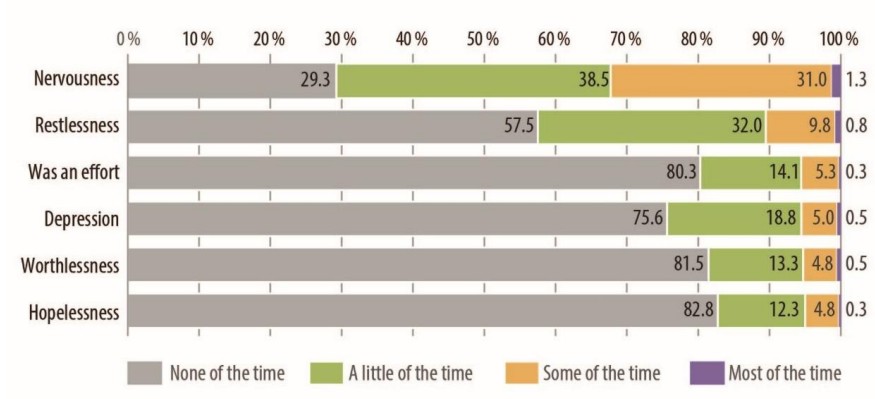


*Figure 3. Symptoms of distress at different scales of K6*

In terms of prevalence, using standard cut-off points, we found that 0.8% of the respondents show severe mental distress, while nearly 19.2 % show moderate levels, and 80% show no indication of mental health distress. Because 0.8% is a small number for the severe mental health group, we combined them with the moderate group. Therefore,
the new proportion becomes 80% and 20% for 'Yes' and 'No', respectively. Then, we generated a new dummy variable for mental health status with the values '1' = 'Yes' and '0'= 'No'. It is used for the binary logistic regression models M-Flood, M-Covid, and M-Mixed.

### 4.3. Mental health in relation to risk factors

Table 3 provides the results of the three logistic models, M-Flood, M-COVID-19, and M-Mixed. The multivariable
logistic regression M-Combined examines the combined impacts of flood and COVID-19 (Table 3).

*Table 3. Results of binary logistic models M-Flood, M-COVID-19 and M-Mixed*

| Domains and direct stressors | Adjusted Odds Ratios (95% conf. interval) | | |
|---|---|---|---|
| | *M-Flood* | *M-Covid* | *M-Mixed* |
| **Flood stressors** | | | |
| Home damage | 1.00 (0.43-2.32) | - | - |
| Livelihood difficulties | **2.39** (1.13-5.04) | - | **2.00** (1.03-3.89) |
| Food and water shortage | 1.08 (0.41-2.81) | - | - |
| Suffered unsanitary condition | 1.02 (0.32-3.27) | - | - |
| Suffered medicine/medical care | 1.52 (0.53-4.41) | - | - |
| Being Evacuated | 0.94 (.42-2.10) | - | - |
| Being rescued | **2.17*** (0.96-4.91) | - | **1.93*** (0.98-3.78) |
| Being seriously injured or ill | 0.82 (0.30-2.25) | - | - |
| Family members/ close friends are injured/ill | 1.87 (0.74-4.71) | - | - |
| Unsure of the safety of family members or close friends | 1.31 (0.64-2.68) | - | - |
| Seeing dead human bodies during or after the flood | **4.93****** (1.98-12.27) | - | **8.67***** (4.03-18.65) |



| | | | |
|---|---|---|---|
| Loss of family members or close friends in the flood | 0.91 (0.28-2.97) | - | - |
| **Covid-19 stressors** | | | |
| .. on your health | - | **4.55\*\*\*** (1.45-14.27) | **2.50\*\*** (1.15-5.45) |
| … someone's health | - | 0.3762519 (0.11-1.25) | |
| ….social maintenance | - | 0.83 (0.43-1.60) | |
| interrupted education | - | **3.03\*\*\*** (1.35-6.80) | **5.67\*\*\*** (1.88-17.11) |
| …Causing household stress | - | 1.59 (0.52-4.80) | |
| **Gender** | **0.61\* (0.32-1.15)** | **0.62\* (0.36-1.05)** | **0.56\* (0.29-1.05)** |
| **Age** | 1.02 (1.00-1.05) | 1.01 (0.99-1.03) | **1.09\*\* (0.86-1.38)** |
| **Education** | 1.10 (0.87-1.40) | 1.04 (0.86-1.27) | 1.03 (1.00-1.06) |

*(\*p-value <0.10, \*\*p-value <0.05, \*\*\*p-value <0.01, \*\*\*\*p-value <0.001. ORs are adjusted with the contextual variables: Gender, Age and Education)*

### 4.3.1 Flood model (M-Flood)

M-Flood shows the ORs of these twelve stressors, three of which are greater than one and statistically significant, including 'Seeing dead human bodies' during or after the flood with the highest ORs of 4.93 (p-value < 0.001), followed by 'Livelihood difficulties' with OR of 2.39 (p-value < 0.05), and lowest one 'Being rescued' with OR is 2.17 (p-value < 0.1). So, among direct flood stressors, 'Seeing dead human bodies' has the highest effect on mental health. No significant effect is found for the rest of the factors, including 'Home damage', 'Food and water shortage', 'Suffered unsanitary condition', 'Suffered medicine/medical care', 'Being Evacuated', 'Being seriously injured or ill', 'Family members/ close friends are injured/ill', 'Unsure of the safety of family members or close friends', and 'Loss of family members or close friends in the flood'. Regarding contextual variables, only 'Gender' is significant in this model with the OR of 0.61 (p-value < 0.1).

### 4.3.2 COVID-19 model (M-COVID-19)

Similar to the flood stressors, direct impacts on mental health distress caused by COVID-19 were explored. The main potential impacts we used are on individual health, someone's health, social maintenance, interrupted education, household stress, and domestic violence. The regression models (see Table 2) show that only two of them are significant, of which the impact of COVID-19 on 'individual health' is more substantial with OR= 4.55 (p-value < 0.01) than 'interrupted education', which has OR = 3.03 (p-value < 0.01). However, 'interrupted education' has a smaller CI, indicating a higher precision of the OR. Like M-Flood, the M-COVID-19 shows that women are more impacted by COVID-19 stressors since 'Gender' has an OR = 0.62 (p-value < 0.1).

### 4.3.3 Mixed stressors (M-Mixed)

M-Mixed was developed from M-Flood and M-COVID-19 by adding three significant flood stressors: 'Seeing dead human bodies', 'Livelihood difficulties' and 'Being rescued'; and two significant COVID-19 stressors: 'impacts on individual health' and 'interrupted education'. All significant variables from M-Flood and M-COVID-19 remain significant in the M-Mixed. The results show the changes in ORs in different ways. For the flood domain, the OR of 'Seeing dead human bodies' increases significantly, from 4.93 to 8.67. While ORs of 'Livelihood difficulties' and 'Being rescued' slightly decrease from 2.37 to 2.00 and 2.17 to 1.93, respectively, the p-value level remains the same





as M-Flood and M-COVID-19. However, the 95%CI of these two variables becomes slightly smaller than M-flood, indicating higher precision.

For the COVID-19 domain, changes in ORs are also shown. The OR of the impact on 'individual health' reduced by two, from 4.55 to 2.50, and the p-value indicates less significance, from < 0.01 to < 0.05, while the 95%CI becomes much smaller from 1.45-14.27 to 1.15-5.45, meaning OR has higher precision. However, 'interrupted education' had

a higher effect, with the OR almost doubled, from 3.03 to 5.67, with the exact p-value, but a much broader 95%CI, from 1.35-6.80 to 1.88-17.11, indicating a less precise OR.

Regarding the contextual variables, it is shown in Table 2 that in M-Flood, M-COVID-19 and M-Mixed, 'Gender' is significant with OR<1, p-values <0.1, and 95%CI does not change much, indicating that women are slightly strongly influenced by flood and COVID-19 in their mental health. Only in M-Mixed, 'Age' becomes significant with OR at

1.087339 (p-value < 0.05, 95%CI: 0.86-1.38). This means some age groups were more affected than others when we mixed all risk factors into one model. There is no effect for 'Education' in all three models.

### 4.3.4 Combined impacts (M-Combined)

Finally, four groups were created using the methods described in Fig. 2 to understand the combined impacts of floods and COVID-19. Group (1) 'NO flood stress and NO COVID stress' has 119 people (37.54%), group (2) 'Flood stress

and NO COVID stress' has 120 people (37.85%), group (3) 'NO flood stress and COVID stress' has 23 people (7.26%), and group (4) 'Flood stress and Covid stress' has 55 people (17.35%). Then, we run a multivariable logistic regression model to find the different mental health suffering (K6) among those four groups, together with contextual variables. The outcome presented in Table 4 reveals interesting information about the combined impact. The OR of the group (1) is the model's base. The ORs were demonstrated with group (1) as the base; group (3), who only faced

COVID-19, had the lowest OR at 2.83 (p-value <0.1). This was followed by group (2), which suffered flooding only, with an OR of 5.47 (p-value <0.001). Notably, in group (4), showing people who experienced both flooding and COVID-19, the OR is highest at 9.67 (p-value < 0.001) (Table 4).

*Table 4. ORs of the combined impacts of Flood and COVID-19 on the mental health status of respondents in four groups*

| N=317 | Odds ratio | P>\|z\| | [95% conf. interval] | |
|---|---|---|---|---|
| No Flood stress & No Covid stress | *(Base)* | | | |
| No flood stress & Covid stress | 2.83 | 0.116 | 0.77 | 10.31 |
| Flood stress & No Covid stress | 5.47 | 0.000 | 2.53 | 11.82 |
| Flood stress & Covid stress | 9.67 | 0.000 | 4.08 | 22.91 |
| *Gender* | *0.52* | *0.028* | *.29* | *0.93* |
| *Age* | *1.02* | *0.076* | *1* | *1.05* |
| *Education* | *1.06* | *0.594* | *.85* | *1.32* |


In model M-Combined, the two contextual variables are significant, including 'Gender' with an OR of 0.52 and p-value < 0.05, and 'Age' with an OR of 1.02 and p-value < 0.1. This indicates that women are at higher risk than men and that the different age groups experience varying levels of effect by the co-occurrence of flood and COVID-19.



**5 Discussion**

**5.1 Prevalence rates**

Our results confirmed that mental health distress exists among respondents who were exposed to various floods and COVID-19 impacts. The prevalence of 20% of respondents with psychological distress demonstrates the effects on vulnerable communities. Even though other research reviewed by Cruz et al. (2020) and Golitaleb et al. (2022) have used different methods and tools for psychological illness assessment, and few of them used mental health distress as

the focus (Butler et al., 2018); our results are consistent with their findings with similar prevalence. As reviewed by Golitaleb et al. (2022), all the relevant studies from 2015 to the middle of 2020 showed that the PTSD prevalence after a flood is high in all age groups. With the research conducted after around two to three years, the typical rate of PTSD is almost 20%. Another review by Cruz et al. (2020) in the UK for the prevalence of depression, anxiety, and PTSD in populations exposed to extreme weather events until December 12[th] 2019; in 17 studies, it was found that within

12 months following extreme weather events, the rate is 19.8% for anxiety, 21.35% for depression, and 30.36% for PTSD.

The findings on COVID-19's impacts on psychological illness in various groups are also confirmed. WHO (2022a) stated that the pandemic has had severe effects on the mental health and well-being of people around the world. UN (2020) released a policy brief on the need for action on COVID-19 and mental health, mentioning the high rate of

mental health distress in some countries, like 35% in China, 60% in Iran and 40% in the US. At the country level, Fernández et al. (2020) discovered distress caused by COVID-19 in Argentina, with participants reporting symptoms of phobic anxiety (41.3%), anxiety (31.8%), depression (27.5%), and general distress (27.1%).

Regarding similar studies on Vietnam, Duong et al. (2020) surveyed 1385 respondents and found that 36% of them experienced psychological distress, 24% depression, 14% anxiety, and 22% stress. Hung et al. (2024) conducted a

cross-sectional study among 125 COVID-19 patients in a centralized quarantined Ho Chi Minh City community. They revealed that the prevalence of depression, anxiety, and stress among patients with COVID-19 was 14%, 21%, and 20%, respectively. Recently, after 4 years, Hoa et al. (2024) found that among 1596 participants in Northern Vietnam, the prevalence of depression, anxiety, sleep disturbance, and cognitive impairment was 9%, 17%, 23%, and 6%, respectively. In Thua Thien Hue, we found one relevant research by Tran et al. (2024), which explored the impact of

the COVID-19 pandemic on returnee migrant workers' income, psychological well-being, and daily life expenses. Their results revealed that reduced income increases the stress of affected people.

From all the mentioned results, our research fills in the research gap on the mental health distress of flood and COVID-19 victims, and the 20% prevalence result is comparable to that of other research. Therefore, it contributes to an overall picture of mental health issues in Vietnam and also points out the more vulnerable groups and areas for mental health

distress. Since then, it has drawn public concern, and the attention of policymakers for supporting policies and action plans to reduce the psychological impacts of simultaneous disasters and pandemics.

**5.2. Risk factors/stressors**

Various types of research have shown that floods have profound effects on the mental health of affected people, including frequently flooded areas and developing countries (Asim et al., 2019; Callender et al., 2022; Cruz et al.,

2020; Ede et al., 2022; Fitzgerald et al., 2020; WHO, 2011). Our M-Flood and M-COVID-19 results find the most significant direct stressors that affect psychological illness caused by floods or the COVID-19 pandemic. Three out of twelve flood factors are significant in M-Flood, including 'Seeing dead human bodies' with the highest OR,



followed by 'Livelihood difficulties' and 'being rescued'. These three stressors are mentioned or analyzed in other studies by Makwana (2019), Abass et al. (2022) and Dai et al. (2016), indicating that the psychological vulnerabilities

of the sufferers may be followed by displacement of the family, death of a loved one, and socio-economic loss, etc. According to NeuroLaunch (2024), the long-term consequences of seeing dead bodies could be PTSD, depression, persistent sadness and helplessness, which have profound impacts on individual lives, relationships, and daily activities. Chapple and Ziebland (2010) proved that interviewed bereaved relatives who saw dead bodies had significantly higher levels of distress and anxiety than those who did not.

Additionally, Tunstall et al. (2006) found that evacuation or rescue and disruption could add more stress to the mental health of affected people. Lamond et al. (2015) added that moving and financial constraints may cause severe mental health issues. These findings raise concerns about livelihood support and rescue planning. Notably, managing the dead remains of individuals who have died in disasters is an important issue. There are some guidelines for this situation, but only to help first responders ensure that the dead are treated (with dignity) and their subsequent identification and

for outbreak prevention (ICRC, 2018; WHO, 2019). In our case study, seeing dead bodies directly or indirectly is associated with the dignity of a dead person and the distress of survivors. Looking back at 2020, some photos of search areas with bodies were circulated in the mass media and on social media. This should be managed differently because it could lead to profound impacts on the mental health of relatives and other viewers.

Interestingly, we found interaction among stressors from the M-Mixed when they were put together in one model. It

is shown in Table 3 that in both domains of flood and COVID-19, among five direct stressors, including 'Seeing dead human bodies', 'Livelihood difficulties', 'Being rescued', 'Covid-19 impact on individual health', and 'interrupted education', two of them had more potent effects on mental health distress with much higher ORs, which are: 'seeing dead human bodies' and 'interrupted education', while three other stressors had decreased ORs. This situation was explained by Schneiderman et al. (2005), that multiple facets of stress can work together and be more potent than a

single facet. The association of psychosocial stressors with illness depends on the types, numbers, and periods of the stressors (Schneiderman et al., 2005). It reminds us of a comprehensive assessment of mental health that considers all potential risk factors in a given situation of co-occurrence, allowing for a more concise focus on support for the victims.

### 5.3. Combined mental health impacts

A key question of the current paper is whether there is a cumulative impact from the two disasters. Our findings clearly reveal that the combined impacts of flooding and COVID-19 on mental health distress are profound in the model M-Combined (Table 4). It shows the highest OR for those who suffered COVID-19 and floods in 2020, compared to flood victims and COVID-19 patients. These results match the few other studies on the combined effects of the coincidence of disasters such as floods and COVID-19 on psychological distress. As concluded by Izumi and Shaw

(2022), the co-occurrence of COVID-19 and natural hazards had extensive and compounding impacts and challenges on the mental health status of affected people. Another example is Callender et al. (2022), who examined the cumulative effects of the flood caused by Hurricane Harvey, along with income loss due to COVID-19 in the US. They concluded that multiple crises have joint impacts on mental health and well-being. In particular, their research found that for those whom Harvey severely affected, the odds ratio of having more severe anxiety during the pandemic

is 5.14 (4.02-6.58) times greater than among registrants for whom Harvey had no meaningful impact. In another research by Podubinski and Glenister (2021), it was similarly revealed that affected people have higher stress



symptoms associated with having suffered a disaster in addition to COVID-19. So, our findings on the impacts of floods and COVID-19 on the psychological distress of affected individuals are feasible and consistent with previous studies. Furthermore, from our case study, we have valuable contributions to examining the co-effects of floods and the COVID-19 pandemic on mental distress, such as screening the prevalence, identifying the risk factors and their interaction, exploring the need for support from at-risk groups and suggesting prevention solutions.

**6 Conclusion**

This research was conducted in Thua Thien Hue province, Central Vietnam, to examine the impacts of flooding and COVID-19 on the mental health of victims. It highlighted the combined impacts of these multiple risks in 2020. KOBO toolbox was used to collect 400 face-to-face surveys in the two communes, focusing mainly on the K6 scale screening tool for the mental health status of affected people and the direct flooding and COVID-19 stressors that local people suffered. Binary logistic regressions and multivariable regression models were used to predict the magnitude of the influence of risk factors on the dependent variable, in this case, the mental health status of respondents via ORs. The research findings align with other relevant studies and make novel contributions to the research topic, with the interesting result of the combined impact of multiple risks. The K6 scale results confirm that psychological distress exists in the affected communities. Concerning flooding, 'livelihood difficulties', 'being rescued' and 'seeing dead human bodies' are three significant direct stressors that affect the mental health status of affected people, whereas 'individual health' and 'interrupted education' are the two main risk factors of COVID-19. These five stressors significantly varied when mixed in one model, proving that they interact with each other. Some of them have higher ORs, while others have lower ORs, compared to the models for flood or COVID-19 only. Moreover, the combined impacts of flood and COVID-19 on the psychological illness of victims proved to be significant. The M-Combined model, which compared groups with different exposure to flooding and/or COVID stressors, shows us the highest ORs of those who suffered flood and COVID-19 impacts in 2020.

These findings help address public health problems resulting from multiple risks rather than focusing on a single risk. First, it draws the attention of relevant stakeholders to a systematic mental health assessment and care service for vulnerable groups and areas, which are still limited in developing countries. Second, the findings highlight the need for support policies and action plans to reduce the psychological impacts of the coincidence of disasters and pandemics. It is necessary to provide additional support to at-risk communities. Lastly, it suggests that some interventions or solutions need to be carefully implemented during and after disasters to prevent or mitigate mental health distress. For example, human remains should be well managed, not only for outbreak prevention but also for protecting the dignity of the deceased and preventing additional distress for the surviving dependents. Rescue plans need to be rehearsed and well-communicated in at-risk communities. Other systematic interventions causing large-scale effects, like school closures, must be carefully assessed. For future research, there is a need for more investigation on this topic in different areas and groups to better understand the cumulative impacts on mental health by the coincidence of disasters for more effective response and prevention activities. It will be helpful to have more studies on the need for support, solutions, and interventions, such as setting up public health infrastructure for psychological assistance.





**Appendices**

**Appendix 1. List of most relevant articles**

| No | Author | Article Title | Objective | DOI |
|---|---|---|---|---|
| 1 | Callender et al., 2022 | Economic and mental health impacts of multiple adverse events: Hurricane Harvey, other flooding events, and the COVID-19 pandemic | Objectives: To assess the economic and mental health impacts of COVID-19 in the presence of previous exposure to flooding events. | 10.1016/j.envres.2022.114 020 |
| 2 | Agyapong et al., 2021 | Mental Health Impacts of Wildfire, Flooding and COVID-19 on Fort McMurray School Board Staff and Other Employees: A Comparative Study | This study aimed to compare the mental health of the school board and other Fort McMurray employees affected by the 2016 wildfires, the 2019 COVID-19 pandemic, and the 2020 floods. | 10.3390/ijerph19010435 |
| 3 | Podubinski & Kristen, 2021 | The Pandemic Is Not Occurring in a Vacuum: The Impact of COVID-19 and Other Disasters on Workforce Mental Health in Australia | This study aimed to provide insight into the mental health of Australian workers during the initial COVID-19 outbreak, with an additional focus on whether previous disaster exposure and impact from that disaster is a risk factor for increased psychological distress. | 10.1017/dmp.2021.238 |
| 4 | Rocha et al., 2021 | Typhoons During the COVID-19 Pandemic in the Philippines: Impact of a Double Crises on Mental Health | This article aims to address the effects of natural disasters on the mental health of Filipinos during the COVID-19 pandemic. | 10.1017/dmp.2021.140 |
| 5 | Agyapong et al., 2022 | Cumulative trauma from multiple natural disasters increases mental health burden on residents of Fort McMurray | This article assesses if the number of traumatic events experienced by residents of Fort McMurray correlates with the prevalence and severity of mental health issues experienced. | 10.1080/20008198.2022.2 059999 |
| 6 | Flood, McFadden & Shepherd, 2022 | The impact of COVID-19 on the mental health of radiography staff and managers in Northern Ireland, UK: The radiography managers' perspective | This study explores radiography managers' perceptions regarding the impact of the COVID-19 pandemic on the mental health of themselves and their staff. | 10.1016/j.radi.2022.06.011 |
| 7 | Mugha et al., 2021 | Psychological impact of the third wave of covid-19 and infodemics on mental health of medical teachers of Islamabad, Pakistan | Objective: To determine mental health problems such as anxiety, depression, and cognitive-behavioural changes in medical teachers due to a sudden rise in COVID-19 cases along with a flood of social media traffic, mostly misinformation. | |
| 8 | Zhai &Lange, 2021 | The Influence of Covid-19 on Perceived Health Effects of Wetland Parks in China | This study explores the public's perception of the health effects of visiting wetland parks and the impact of the pandemic on the perception. | 10.1007/s13157-021-01505-7 |
| 9 | Agyapong et al., 2023 | Mental Health Impacts of Wildfire, Flooding and COVID-19: on educators: A Comparative Study | This study aimed to compare employees of the school board and other employees of Fort McMurray with respect to the impact the 2016 wildfires, the 2019 COVID pandemic, and the 2020 floods had on their mental health. | 10.1192/j.eurpsy.2023.201 3 |



| 10 | Zhang & Jia, 2023 | When fate hands you lemons: A moderated moderation model of bullying victimization and psychological distress among Chinese adolescents during floods and the COVID-19 pandemic | This study examined the moderating effects of neuroticism and negotiable fate on the relationship between bullying victimization and psychological distress among Chinese adolescents. This study included participants who experienced floods and COVID-19 simultaneously in 2021. | 10.3389/fpsyg.2023.101040 08 |
|----|----|----|----|----|
| 11 | Shakespeare-Finch et al.,2020 | COVID-19: An Australian Perspective | In Australia, the pandemic came on the back of the largest bushfire season the country had seen, which followed a sequence of climatic disasters involving drought, cyclones and floods. This study highlights the mental health risk that may arise from increased sedentary behavior with the introduction of lockdown and physical distancing measures. Also, it outlines the potently valuable role of drawing on salutogenic models including resilience and posttraumatic growth research for individual and broader community level need. | 10.1080/15325024.2020.1 780748 |
| 12 | Liang et al., 2023 | Latent profiles of psychological status among populations cumulatively exposed to a flood and the recurrence of the COVID-19 pandemic in China | The current study aims to identify the latent profiles of psychological status and acceptance of change among Henan residents who have been cumulatively exposed to floods and the COVID-19 pandemic. | 10.1016/j.ijdrr.2022.103520 0 |
| 13 | Izumi & Shaw, 2022 | A multi-country comparative analysis of the impact of COVID-19 and natural hazards in India, Japan, the Philippines, and USA | This study investigated the impact of COVID-19 on disaster response and recovery from various types of hazards with regard to preparedness, evacuation, volunteering, early recovery, awareness and knowledge of different types of hazards, and preparedness capacity development. This study targets hazards such as Cyclone Amphan in India, the Kumamoto flood in Japan, Typhoon Rolly in the Philippines, and the California wildfires in the U.S. | 10.1016/j.ijdrr.2022.102899 9 |
| 14 | Sheehan, 2022 | 2021 Climate and Health Review - Uncharted Territory: Extreme Weather Events and Morbidity | This review summarizes data for 30 major EWEs of 2021 and, based on the epidemiological literature, discusses morbidity-related exposures for four hazards that marked the year: wildfire smoke, extreme cold and power outages, extreme precipitation-related flooding, and drought. | 10.1177/00207314221082 452 |
| 15 | Jing & Katz, 2021 | An update on psychotic spectrum disorders and disasters | The aim of this study is to review the recent literature on disasters' impact on the course of psychotic spectrum disorders (PSDs) and how people with PSD fare during a disaster, including the effects of COVID-19. | 10.1097/YCO.0000000000 000700 |



| 16 | Feng et al., 2023 | The workload change and depression among emergency medical staff after the open policy during COVID-19: a cross-sectional survey in Shandong, China | This study investigates the workload change, prevalence, and associated factors for depression symptoms among emergency medical staff after the policy adjustment. Open policies were associated with higher PHQ-9 scores for those from grade-B tertiary hospitals. Hospital administrators should reinforce the importance of targeted emergency medical staff support during future outbreaks. | 10.3389/fpubh.2023.1281787 |
|---|---|---|---|---|
| 17 | Kumar & Somani, 2020 | Dealing with Coronavirus anxiety and OCD | The world is reeling under the crisis caused by coronavirus disease (COVID-19); print, electronic and social media are flooded with numerous advisories issued by governments and other national & international agencies. While all this is being done with the best of intentions to contain the spread of this viral disease, this is causing a significant negative impact on the mental health of people, especially persons with obsessive-compulsive disorder with fear of contamination and excessive washing of hands. | 10.1016/j.ajp.2020.102053 |
| 18 | Tran et al., 2023 | Interruptions to HIV Care Delivery During Pandemics and Natural Disasters: A Qualitative Study of Challenges and Opportunities From Frontline Healthcare Providers in Western Kenya | The goal of this study was to understand the impact of the COVID-19 pandemic and recent flooding disasters on HIV care delivery in western Kenya. | 10.1177/23259582231152041 |

**Appendix 2. Variable summary included for analysis**

| Variable name | Variable description | Variable descriptive statistics |
|---|---|---|
| **Dependent variable:** K6 mental distress scale Responding to the questions asking about: How have respondents been feeling during the past 30 days? | | |
| Nervous | Coded: from 0=None of the time, 1=A little of the time, 2=Some of the time, 3=Most of the time, 4=All of the time, and 99=Don't know/No answer | N= 400 Min=0, Max=3 Mean=1.04 |
| Hopeless | Coded: from 0=None of the time, 1=A little of the time, 2=Some of the time, 3=Most of the time, 4=All of the time, and 99=Don't know/No answer | N= 400 Min=0, Max=3 Mean=0.23 |
| Restless | Coded: from 0=None of the time, 1=A little of the time, 2=Some of the time, 3=Most of the time, 4=All of the time, and 99=Don't know/No answer | N= 400 Min=0, Max=3 Mean=0.54 |
| Depressed | Coded: from 0=None of the time, 1=A little of the time, 2=Some of the time, 3=Most of the time, 4=All of the time, and 99=Don't know/No answer | N= 398 Min=0, Max=3 Mean=0.30 |
| Everything was an effort | Coded: from 0=None of the time, 1=A little of the time, 2=Some of the time, 3=Most of the time, 4=All of the time, and 99=Don't know/No answer | N= 396 Min=0, Max=3 Mean=0.26 |
| Worthless | Coded: from 0=None of the time, 1=A little of the time, 2=Some of the time, 3=Most of the time, 4=All of the time, and 99=Don't know/No answer | N= 399 Min=0, Max=3 Mean=0.24 |
| **Flood stressors** | | |



| Part 1: During the previous flood in October 2020 and in the aftermath, did you suffer… | | |
|---|---|---|
| Home damage | Coded: 1=None, 2=A Little, 3=Some, 4=A lot, 5=Extreme, 99=Don't know/No answer | N=324 Min=1, Max=5, Mean=2.69 |
| Livelihood difficulties | Coded: 1=None, 2=A Little, 3=Some, 4=A lot, 5=Extreme, 99=Don't know/No answer | N=324 Min=1, Max=5, Mean=2.67 |
| Food and water shortage | Coded: 1=None, 2=A Little, 3=Some, 4=A lot, 5=Extreme, 99=Don't know/No answer | N=324 Min=1, Max=5, Mean=2.29 |
| Unsanitary condition | Coded: 1=None, 2=A Little, 3=Some, 4=A lot, 5=Extreme, 99=Don't know/No answer | N=324 Min=1, Max=5, Mean=2.10 |
| Medicine/medical care | Coded: 1=None, 2=A Little, 3=Some, 4=A lot, 5=Extreme, 99=Don't know/No answer | N=324 Min=1, Max=5, Mean=1.90 |
| Part 2: Please answer with "yes" or "no" if you had the following experiences during the previous flood in October 2020… | | |
| Evacuated | Coded: 1=Yes, 0=No, 99=Don't know-no answer | N=323 Min=0, Max=1, Mean=0.69 |
| Rescued | Coded: 1=Yes, 0=No, 99=Don't know-no answer | N=318 Min=0, Max=1, Mean=0.25 |
| Injured/ill | Coded: 1=Yes, 0=No, 99=Don't know-no answer | N=323 Min=0, Max=1, Mean=0.22 |
| Family member/close friend injured/ill | Coded: 1=Yes, 0=No, 99=Don't know-no answer | N=322 Min=0, Max=1, Mean=0.23 |
| Unsured Safety | Coded: 1=Yes, 0=No, 99=Don't know-no answer | N=302 Min=0, Max=1, Mean=0.44 |
| See dead human bodies during/after the flood | Coded: 1=Yes, 0=No, 99=Don't know-no answer | N=323 Min=0, Max=1, Mean=0.19 |
| Loss of close family member / close friend | Coded: 1=Yes, 0=No, 99=Don't know-no answer | N=323 Min=0, Max=1, Mean=0.06 |
| COVID-19 stressors: How did COVID-19 impact on …? | | |
| Your health | Coded: 1=Not at all, 2=A little, 3=Somewhat, 4=A lot, 5=Extremely, 99=Don't know / No answer | N= 400 Min=1, Max=5, Mean=2.05 |
| Somebody else's health | Coded: 1=Not at all, 2=A little, 3=Somewhat, 4=A lot, 5=Extremely, 99=Don't know / No answer | N= 400 Min=1, Max=5, Mean=2.09 |
| Social maintenance | Coded: 1=Not at all, 2=A little, 3=Somewhat, 4=A lot, 5=Extremely, 99=Don't know / No answer | N= 400 Min=1, Max=5, Mean=2.79 |
| Interrupted education | Coded: 1=Not at all, 2=A little, 3=Somewhat, 4=A lot, 5=Extremely, 99=Don't know / No answer | N= 397 Min=1, Max=5, Mean=1.62 |
| Household stress | Coded: 1=Not at all, 2=A little, 3=Somewhat, 4=A lot, 5=Extremely, 99=Don't know / No answer | N= 400 Min=1, Max=5, Mean=1.79 |
| Domestic Violence | Coded: 1=Not at all, 2=A little, 3=Somewhat, 4=A lot, 5=Extremely, 99=Don't know / No answer | N= 399 Min=1, Max=4, Mean=1.03 |
| Contextual variables | | |
| Gender | Coded: 1=Male, 0=Female | N=400, Mean=0.47 |
| Age | Continuous variable | N=400, Min=17, Max=87, Mean=49.4 |
| Education level | Coded: 1=No formal education, 2=Primary, 3=Secondary, 4=High School, 5=Techinical Diploma, 6=University | N=400, Min=1, Max=6 Mena= 2.47 |




**Data availability**

Data is available from the authors upon reasonable request.

**Author contributions**

Thi Dieu My Pham: Writing original and final drafts, Research Design, Methodology, Formal analysis,

Investigation, Conceptualization,

Paul Hudson: Writing - review & editing, Methodology, Conceptualization.

Annegret H Thieken: Writing- review & editing, supervising, and Funding acquisition.

Philip Bubeck: Writing - review & editing, Methodology, Conceptualization, supervising, and Funding acquisition.

**Competing interests**

The contact author has declared that none of the authors has any competing interests.

**Acknowledgements**

We also thank the Centre for Social Research and Development for carrying out the data collection activities.
Special thanks go to Beate Swanson for proofreading the article and Ute Dolezal and Christina Schmidt for editing
the figures in this article. Lastly, we thank the local authorities and participants in TTH for their support.




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
