# Peer review of "Understanding the combined mental health impacts of flooding and COVID-19 in Hue City, Central Vietnam"

_EGUsphere, 2025_

## Referee Comment (RC1)

Thank you for offering an opportunity to review a manuscript that investigated the compound impact of COVID-19 and flood on the mental health of a flood-prone area in Vietnam. The manuscript is well-written and structured, with methodological integrity. However, there are some queries the authors need to address.

**Introduction**

In page 2, line 55, the statement 'while the flood ...... and under-evaluated' and the following references are misleading, as the authors cited a paper prior to the COVID-19. It would be a good choice if the authors cite Ogunbode et al. (2019) after 'While the flood impacts on mental health are considerable and long-lasting.'

Otherwise, the *Introduction* section is well-written, documenting relevant articles in both global and regional contexts.

**Methodology**

- 2. The authors clearly documented the simultaneous occurrence of floods and COVID-19 in the selected study areas. They specifically mentioned using the K6 scale to measure distress among the residents of Hue City. Importantly, the K6 scale is designed to assess an individual's mental health, specifically measuring distress experienced over the past 30 days (see Kessler, R. C., Barker, P. R., Colpe, L. J., Epstein, J. F., Gfroerer, J. C., Hiripi, E., . . . Zaslavsky, A. M. (2003). Screening for serious mental illness in the general population. *Archives of General Psychiatry*, 60(2), 184–189. doi: 10.1001/archpsyc.60.2.184). The authors examined the mental health of residents in Hue City concerning incidents that occurred over two years ago. This raises a question about how they addressed the potential recall bias among participants and whether they considered individuals who may have relocated to study areas after 2022 from other parts of the country. Additionally, the authors did not specify any inclusion or exclusion criteria for the population studied.
- 3. The authors stated they interviewed 400 household heads using systematic random sampling from two communes; however, it is unclear if this sample size adequately represents the study areas.
- 4. The authors are requested to mention the reliability of K6 in their study.
- 5. The authors are requested to provide any relevant articles that support merging 'moderate' and 'severe' distress into the 'Yes' category. If they choose not to merge 'no' and 'moderate' distress into the 'no or moderate' distress category, they must provide an explanation for this decision.

The authors are requested to address or clarify the aforementioned issues in the methodology.

---

## Author Comment (AC1)

**Response letter to Reviewer 1.**

Dear Reviewer,
Thank you for your time and effort in reviewing the manuscript. We appreciate your positive evaluation of our paper and the comments and suggestions that will help to improve our manuscript further. We would like to address the comments below (in italics) and explain in our detailed point-by-point response how we will incorporate these changes into the revised version of our manuscript.

Kind regards on behalf of all authors,
*Thi Dieu My Pham*

Thank you for offering an opportunity to review a manuscript that investigated the compound impact of COVID-19 and flood on the mental health of a flood-prone area in Vietnam. The manuscript is well-written and structured, with methodological integrity. However, there are some queries the authors need to address.

*Thank you for the overall very positive feedback.*

1. In page 2, line 55, the statement 'while the flood …… and under-evaluated' and the following references are misleading, as the authors cited a paper prior to the COVID-19. It would be a good choice if the authors cite Ogunbode et al. (2019) after 'While the flood impacts on mental health are considerable and long-lasting.'
Otherwise, the Introduction section is well-written, documenting relevant articles in both global and regional contexts.

*We thank the reviewer for this recommendation. We changed this citation for clarification. In the revised manuscript, it will read as: "The co-occurrence of floods and COVID-19 is a typical multi-hazard event (Gill et al., 2022; Simonovic et al., 2021). While the flood impacts on mental health are considerable and long-lasting (Ogunbode et al., 2019), a systematic review of health sector responses to the coincidence of disasters and COVID-19 is underreported and under-evaluated (Sedighi et al., 2021)."*

Gill, J.C., Duncan, M., Ciurean, R., Smale, L., Stuparu, D., Schlumberger, J, de Ruiter M. (2022). *P. 2022. MYRIAD-EU D1.2 Handbook of Multi hazard, Multi-Risk Definitions and Concepts.: H2020 MYRIAD-EU Project, grant agreement number 101003276.*

Ogunbode, C. A., Böhm, G., Capstick, S. B., Demski, C., Spence, A., & Tausch, N. (2019). The resilience paradox: flooding experience, coping and climate change mitigation intentions. *Climate Policy, 19*(6), 703–715.

Sedighi, T., Varga, L., Hosseinian-Far, A., & Daneshkhah, A. (2021). Economic Evaluation of Mental Health Effects of Flooding Using Bayesian Networks. *International journal of environmental research and public health, 18*(14).

Simonovic, S. P., Kundzewicz, Z. W., & Wright, N. (2021). Floods and the COVID-19 pandemic-A new double hazard problem. *WIREs. Water, 8*(2), e1509.

**Methodology**

1. The authors clearly documented the simultaneous occurrence of floods and COVID-19 in the selected study areas. They specifically mentioned using the K6 scale to measure distress among the residents of Hue City. Importantly, the K6 scale is designed to assess an individual's mental health, specifically measuring distress experienced over the past 30 days (see Kessler, R. C., Barker, P. R., Colpe, L. J., Epstein, J. F., Gfroerer, J. C., Hiripi, E., . . . Zaslavsky, A. M. (2003). Screening for serious mental illness in the general population. *Archives of General Psychiatry*, *60*(2), 184–189. doi: 10.1001/archpsyc.60.2.184). The authors examined the mental health of residents in Hue City concerning incidents that occurred over two years ago. This raises a question about how they addressed the potential recall bias among participants and whether they considered individuals who may have relocated to study areas after 2022 from other parts of the country. Additionally, the authors did not specify any inclusion or exclusion criteria for the population studied.

*Thank you for raising this point, which allows us to explain the method, the study site, and the sample in more detail.*

*Firstly, as the reviewer mentioned, the K6 scale indeed captures the last 30 days, and it is well known that both flooding and COVID-19 can have a long-lasting effect on mental health. The K6 is not significantly affected by the recall period and can help identify the significance of changes in psychological distress over time and evaluate the effectiveness of interventions for psychological distress (Chilver et tal., 2023; Unchiba et al., 2023). Regarding stressors potentially affected by recall bias, we argue that COVID-19 and the flood event were highly impactful for respondents, and, based on our experience, we did not observe problems with respondents' recall.*

*Secondly, in the first part of the questionnaire, we asked about the 2020 flood experience in the two selected communes. The results show that only 2.9% (4 people) of the respondents in Hai Duong commune and 1.1% (2 people) in Quang Loi commune were not affected by the flood*

*event in that year. This means the situation raised by the reviewer (moving in of respondents) may potentially be the case for these 6 out of 400 respondents, and is therefore, neglectable.*

Uchida H, Kuroiwa C, Ohki S, Takahashi K, Tsuchiya K, Kikuchi S, Hirao K (2023). Assessing the Smallest Detectable Change of the Kessler Psychological Distress Scale Score in an Adult Population in Japan. Psychol Res Behav Manag. DOI: 10.2147/PRBM.S417446. PMID: 37465046; PMCID: PMC10351679.

Chilver MR, Burns RA, Botha F, Butterworth P (2023) Comparing estimates of psychological distress using 7-day and 30-day recall periods: Does it make a difference? PLoS ONE 18(12): e0295535. https://doi.org/10.1371/ journal.pone.0295535

2. The authors stated they interviewed 400 household heads using systematic random sampling from two communes; however, it is unclear if this sample size adequately represents the study areas.

*We would like to clarify that the survey sites and the number of respondents were defined within the project scope, targeting coastal and lagoon communities that are affected by floods each year. Regarding the sampling size, we think that 400 people is a reasonable number to represent the population, in accordance with the formula by Taro Yamane (1967), which applies to both infinite and finite populations. Since there is only an estimate of about 500,000 people in the lagoon area (hue.gov.vn), we used both formulas.*
*First, the formula for an infinite population (either uncountable or too large to ever count or measure):*

$$n = Z^2 \times \frac{p \times (1-p)}{e^2}$$

*In which:*
***n**: the sample size to be determined.*
***Z**: the Z-value obtained from the standard normal distribution table based on the selected confidence level. Commonly, a 95% confidence level is used, corresponding to **Z = 1.96**.*
***p**: the estimated proportion of success in the population. Typically, **p = 0.5** is chosen because it maximizes the product **p(1 - p)**, ensuring a conservative (i.e., safe) sample size estimate.*
***e**: the allowable margin of error. Three commonly used error levels are: ±0.01 (1%), ±0.05 (5%), and ±0.1 (10%), with **±0.05** being the most widely used.*
*The result is:*

$$n = 1.96^2 \times \frac{0.5 \times (1-0.5)}{0.05^2} = 384.16$$

*This means we need at least 384 participants for our study.*

*For the finite population (that is, countable and has a known or measurable number of members), we used this formula:*

$$n = \frac{N}{1 + N \times e^2}$$

*In which:*

- ***n**: the sample size to be determined.*
- ***N**: the population size (total number of units in the population).*
- ***e**: the allowable margin of error. Commonly used error rates are: ±0.01 (1%), ±0.05 (5%), and ±0.1 (10%), with ±0.05 being the most used.*

*It is estimated that 500,000 people live in the surrounding lagoon area of Hue city (hue.gov.vn), we have:*

$$n = \frac{500{,}000}{1 + 500{,}000 \times 0.05^2} = 400$$

*Based on the results of both formulas, our sample size adequately represents the targeted population.*

*Also, the sample characteristics match the city's descriptive statistics reasonably. Except for income, which is sometimes sensitive to ask, gender, age, and education are relatively homogeneous with the city's descriptive statistics and reflect the local context of the lagoon area (TTH Statistic Office, 2022). Because of random sampling, we have tried to eliminate any systematic connection between sampling bias and our dataset.*

TTH Statistic Office (2022). *Thua Thien Hue Statistical Yearbook,* from Thua Thien Hue Statistic Office: https://drive.google.com/file/d/1YA7gIxKr4TjMcQWDphL9UpKpFrWGmrXd/view.

3. The authors are requested to mention the reliability of K6 in their study.

*We agree with this suggestion and will add this component to the article in the introduction of the K6 as follows: According to Wojujutari and Idemudia (2024), who worked on 'A Reliability Generalization Meta-Analysis of Kessler Psychological Distress Scale (K-10 and K-6), the scales were adapted into multiple languages, including English, Chinese, Swahili, Farsi, Indonesian, Japanese, Hindi, and Portuguese, reflecting their global applicability and adaptability. For the K-6, their results revealed high internal consistency (mean α = 0.84, 95% CI [0.80, 0.88]).*

*Reliability varied across populations and languages. The K-6 scale showed a high reliability among outpatients (α = 0.89) and the general population (α = 0.87). The authors concluded that the K6 is a reliable tool for measuring psychological distress in both general and clinical groups. Its high reliability and adaptability make it valuable in clinical practice and research. It is recommended to use and adapt global mental health assessments, with attention to cultural and language considerations. In addition, in the article 'Screening for Serious Mental Illness in the General Population' by Kessler et al. (2003), the authors showed that K6 is robust for screening for Serious Mental Illness and concluded that it is crucial for clinical studies and clinical epidemiology.*

Kessler RC, Barker PR, Colpe LJ, et al (2003). Screening for Serious Mental Illness in the General Population. *Arch Gen Psychiatry. 60(2):184–189. DOI:10.1001/archpsyc.60.2.184*

Wojujutari AK, Idemudia ES (2024). Consistency as the Currency in Psychological Measures: A Reliability Generalization Meta-Analysis of Kessler Psychological Distress Scale (K-10 and K-6). *Depress Anxiety. DOI: 10.1155/2024/3801950. PMID: 40226709; PMCID: PMC1191909.*

4. The authors are requested to provide any relevant articles that support merging 'moderate' and 'severe' distress into the 'Yes' category. If they choose not to merge 'no' and 'moderate' distress into the 'no or moderate' distress category, they must explain this decision. The authors are requested to address or clarify the issues mentioned above in the methodology.

*We thank the reviewer for this suggestion, and we would like to clarify our decision as follows: The widely used Kessler K6 non-specific distress scale screens for severe mental illness, defined as a K6 score ≥ 13, estimated to afflict about 6% of US adults. The K6, as currently used, fails to capture individuals struggling with more moderate mental distress that nonetheless warrants mental health intervention." They provide a cut-off for 'moderate' mental distress, which we use and which is appropriate for your research question. Since the severe group is too small, the prevalence shows only 0.8%. It is too small to analyze as a group. Combining this group with the moderate helped refine the analysis by comparing the presence of stress experience with no experience at all, providing a more precise comparison (Perez-Valero et al., 2021). Prochaska et al. (2012) also indicated that it is necessary to use K6 to detect, examine, and quantify the correlates of moderate distress, given its clinical relevance.*

Eduardo Perez-Valero, Miguel A. Lopez-Gordo, Miguel A. Vaquero-Blasco (2021). EEG-based multi-level stress classification with and without smoothing filter. *Biomedical Signal Processing and Control. Volume 69, 102881, ISSN 1746-8094,* https://doi.org/10.1016/j.bspc.2021.102881.

US Centers for Disease Control and Prevention *(https://www.cdc.gov/mental-health/about/index.html)*

Prochaska, J. J., Sung, H.-Y., Max, W., Shi, Y., & Ong, M. (2012). Validity study of the K6 scale as a measure of moderate mental distress based on mental health treatment need and utilization. *International Journal of Methods in Psychiatric Research, 21(2), 88–97. https://doi.org/10.1002/mpr.1349*